# Hardening Slurries with Fluidized-Bed Combustion By-Products and Their Potential Significance in Terms of Circular Economy

**DOI:** 10.3390/ma14092104

**Published:** 2021-04-21

**Authors:** Zbigniew Kledyński, Paweł Falaciński, Agnieszka Machowska, Łukasz Szarek, Łukasz Krysiak

**Affiliations:** Faculty of Building Services, Hydro and Environmental Engineering, Warsaw University of Technology, Nowowiejska St. 20, 00-653 Warsaw, Poland; zbigniew.kledynski@pw.edu.pl (Z.K.); pawel.falacinski@pw.edu.pl (P.F.); lukasz.szarek@pw.edu.pl (Ł.S.); lukasz.krysiak@pw.edu.pl (Ł.K.)

**Keywords:** hardening slurry, cement-bentonite slurry, fluidized bed ash, cementitious materials, heavy metal leaching, circular economy

## Abstract

Hardening slurries (water-bentonite-binder mixtures) constitute a well-established material used broadly, i.a., for cut-off walls in civil and water engineering. Although they usually contain Portland cement, similar to common concrete, their properties differ greatly, mostly due to a much higher water content. This characteristic of hardening slurries creates unique opportunities for the utilization of significant quantities of industrial by-products that are deemed problematic in the concrete industry. This article investigates the effect of the addition of by-products of fluidized-bed combustion of hard, brown coal and municipal sewage sludge, as well as ground granulated blast furnace slag, on the properties of slurries. Unconfined compressive strength tests, as well as mercury porosimetry, scanning electron microscopy, and X-ray diffraction analyses were performed. The results suggest that it is possible to design hardening slurry mixes of desired properties, both in liquid and solid state, containing at least 100–300 kg/m^3^ of industrial waste. This includes cement-free slurries based entirely on industrial by-products as binders. In addition, the analyzed slurries exhibited good chemical resistance to landfill eluates, at the same time effectively immobilizing heavy metals. It was concluded that hardening slurry technology can ensure the safe deposition of significant amounts of waste that would be otherwise difficult to manage, thus contributing to the circular economy concept.

## 1. Introduction

Clay mineral aqueous slurries have been used in drilling and civil engineering for centuries. Beginning in the 1960s, however, bentonite-water slurries were enriched with cement binder, which resulted in their transformation into a solid body, owing to cement binding and hardening, after the fluidity period required during trench excavation [1]. These so-called “hardening slurries” can be therefore defined as mixtures of water, binder (usually cement-based), and clay mineral (usually bentonite) with other optional additives [2]. They exhibit thixotropic properties in the liquid state and have the ability to set through chemical bonding. They can be used, i.a., to embed prefabricated elements in the ground, fill voids in a ground substrate, and execute trench cut-off walls [3].

Cut-off walls are vertical barriers used in hydraulic engineering that are embedded in the ground and are spread across a predicted water filtration path. Their task is to limit groundwater flow, e.g., in embankment dams and their subsoil, in levees and to protect excavations. Moreover, they can be applied to prevent pollutants from penetrating into groundwater, e.g., from landfills [4,5]. Slurry trench technology enables the execution of a thin and relatively deep barrier in the ground, with a relatively small scope of earthwork and without the need to erect excavation-protecting structures. Thus, many cases of its use have been reported, e.g., in [6]. The principle of operation of a typical cut-off wall and its execution method (single-phase) are shown in schematic form in Figure 1. Wherever possible, a cut-off wall should reach down to low-permeable soil layers [4].

Hardening slurry technology allows the execution of cut-off walls using the single- or two-phase method. In the first case, the excavation slurry (digging stage) already has a full composition, capable of later bonding (starting from phase (a) in Figure 1), whereas in the two-phase method, the excavation slurry is in the (a) phase in Figure 1, a bentonite–water slurry, while the binder is added only after reaching full depth (at the end of phase (b) in Figure 1). Mixing can be conducted in the excavation, or the excavation slurry can be substituted with the hardening slurry through displacement [4].

The execution technology and the operating nature of ready cut-off walls determine the expected properties of hardening slurries in the liquid state (technological properties) and after hardening (performance properties), respectively. At the “execution” stage, the slurry should support excavation walls through hydrostatic pressure and also seal them, thereby preventing an excessive penetration of liquid into the ground. At the “operating” stage, in contrast, it should primarily be effective in limiting groundwater filtration while maintaining durability [3,4,5].

The technological properties based on [3,4,7,8] and their significance are as follows (the values in parentheses are typical for slurries used in Poland, based on [9]):bulk density: determines the pressure supporting excavation walls and displacement potential in the two-phase method (1.15–1.55 g/cm^3^);viscosity: important in the stages of slurry production, pumping, and trench execution and determines the susceptibility of the slurry to penetration into the ground and to displacement in the two-phase method (35–70 s);water bleed: measure of slurry stability and its segregation tendency (0–6%);structural (gel) strength: associated with slurry thixotropy, which is the sole ability to turn into gel while resting. This gel exhibits a certain shear strength, referred to as “structural”, which limits the sedimentation of slurry components and excavated soil (from 5 to over 30 Pa).

Slurry performance properties include (typical values, based on [9]):hydraulic conductivity (filtration coefficient k): a sufficiently low conductivity is the most important feature of a material in a finished cut-off wall (<10^−8^ m/s after 28 days of curing) [10];compressive strength: important in terms of material durability and maintaining cut-off wall integrity under earth and groundwater pressure conditions. Slurries typically achieve a uniaxial compressive strength of no more than several MPa (above 0.5–1.0 MPa) [11];corrosion resistance: the filtering water might contain pollutants that can react with the components of a hardened slurry. The available test methods involve slurry specimen storage in aggressive solutions or filtration through the specimen using aggressive solutions. Symptoms evidencing susceptibility to aggression are mass loss, strength reduction, and an increasing filtration coefficient [4]. A review by Huang et al. [1] claims that poor resistance to sulfates and acids is a major problem for many common slurry mixes;ability to immobilize heavy metals: the use of industrial waste in the composition of a hardening slurry poses a risk of pollutants (e.g., heavy metals) contained therein being released to groundwater. Immobilization testing involves, e.g., measuring heavy metal content in filtrates originating from the hardening slurry specimen filtration testing [12].

The current primary energy sources in Poland are hard coal and brown coal. Their combustion leads to the formation of approximately 23 million tonnes of slag and fly ash (hereinafter referred to as combustion by-products (CBPs)), almost 60% of which are used in various fields of civil engineering [13]. A distinguishing feature of Poland’s coal-fired power sector is the significant share of fluidized-bed boilers in the industry. These also involve flue gas desulphurization processing. While fluidized-bed boilers are more favorable to the atmosphere, their CBPs exhibit properties less useful in terms of their future application. The high variability of fluidized-bed ash properties and the presence of calcium sorbent reacted to a different degree are important issues hindering the utilization of fluidized-bed ash in the construction industry [14]. These circumstances constitute an obstacle against their wider application in today’s cement and concrete technology.

When looking for possible fluidized-bed CBP applications, attention was drawn to highly aqueous clay mineral slurries. What is considered a major difficulty in concrete (secondary ettringite swelling) becomes an advantage in slurries, since the complex salts present in cement binders can constitute additional sealants as their volume increases during hydration [10]. Therefore, hardening slurries with an addition of CBPs are a potentially attractive construction material and an effective way to manage the by-products of combustion of materials other than coal, e.g., wastes that require thermal treatment, such as sewage sludge, municipal waste, etc. Since single-phase slurries must combine beneficial liquid state and hardened state features, the two-phase method has a greater potential in terms of CBPs utilization, as it enables a more flexible shaping of slurry composition and properties, e.g., a relatively high bulk density. Such an approach should lead to the maximized utilization of combustion by-products within the material, without sacrificing its essential performance properties, including high watertightness and corrosion resistance.

Past research provides well-documented proof of the feasibility of using granulated blast furnace slag and fly ash from coal fired power plants in hardening slurries [1,2], but CBPs from fluidized-bed furnaces, especially those burning non-coal fuels, have been comparatively less thoroughly tested in this field [7,10,15].

The aim of this article was to research hardening slurries with CBPs obtained from the fluidized-bed combustion of coals and sewage sludge, as well as cement-free slurries where a mixture of ground blast furnace slag (steel smelting by-product) and brown coal fluidized-bed combustion fly ash play the role of a binder. The studies are focused on slurries dedicated for cut-off walls and the material properties that are fundamental in this respect, i.e., watertightness after hardening, resistance to the chemical aggression of a groundwater environment, and the ability to immobilize heavy metals.

The significance of the obtained results is highlighted in terms of efficiently enriching hardening slurries with certain kinds of CBPs that exhibit properties hindering or preventing their use by methods well-established for the typical CBPs. The presented research on hardening slurries is discussed in the light of contemporary challenges arising from excessive waste generation. In this perspective, combustion by-products become a component of hardening slurries not in order to improve the technical or economic parameters of the slurries, but because the slurries then become a method of disposing of as much waste as possible, while maintaining the required technological and performance properties. Such an approach is parallel with polymer-modified concrete technology [16]. In the context of hardening slurries, it opens up additional, process-related capacities of the circular economy, among others, in terms of the concept of zero-waste coal energy [17], especially in the face of hard-to-manage fluidized-bed combustion waste.

## 2. Materials and Methods

### 2.1. Recipes of Tested Slurries

The compositions of the tested hardening slurries (A–G) are given in Table 1.

A hardening slurry component that is predominant in terms of mass and volume is water. Water that satisfies requirements set for the production of concrete mixes is typically used to manufacture slurries [18]. The component that gives the hardening slurry its thixotropic properties is bentonite, a strongly swelling sedimentary rock that consists primarily of montmorillonite—a mineral that, due to its package-like structure, is characterized by developed specific surface area of approx. 800 m^2^/g and high water absorption and swelling capacity [19]. Owing to its sorptive properties, bentonite can also be used to adsorb heavy metals [20].

The binder most frequently employed in hardening slurries is CEM I Portland cement. Owing to its composition, as well as curing and operating conditions, the class of applied cement does not matter; there is no need to use cements with accelerated early strength increment, low alkali content, and reduced heat of hydration. In the case of a hardening slurry under chemical aggression conditions, it is recommended to use multi-component Portland cement, composite cement, CEM III blast furnace cements, or slag binders enriched with fly ash [4,21,22]. A brief description of additional hardening slurry components is presented below.

### 2.2. Fluidized-Bed Combustion Fly-Ash

Due to the differences in the combustion process relative to conventional coal-fired boilers, fluidized beds produce a different type of CBPs. The differences compared to well-studied conventional combustion ash are as follows.

Fluidized-bed fly ash does not contain spherical and vitrified grains; however, it is similar to pulverized coal furnaces in terms of grain size distribution. Furthermore, the grains in this type of ash have a much more developed specific surface area. Fluidized-bed fly ash properties depend on the type of combusted coal, sulphur content in the fuel, type of used sorbent, type of boiler, combustion method and temperature, and degree of oxidation of flue gas desulphurization products [23].

Fluidized-bed boiler fly ash is characterized by high chemical composition variability (Table 2), which influences its application possibilities within the construction industry. Fluidized-bed ash, compared to pulverized coal boiler ash, contains more sulphates (SO_3_) and total calcium oxide (CaO). Loss on ignition, as the total content of non-combusted coal and losses originating from calcium carbonate decomposition, is higher than in the case of pulverized coal ash [14].

In terms of phase composition, fluidized-bed coal combustion ashes contain a small amount of the glassy phase (which results from the relatively low temperature (~850 °C) within the bed) and semi-amorphous products of a clayey substance (gangue). The last are aluminium silicates (illite) with relatively large specific surface areas. This has influence on the pozzolanic activity of the ashes. These ashes also contain free calcium oxide and Ca(OH)_2_ as the product of its hydration, as well as anhydrite derived from the desulphurization process, and unreacted sorbent in the form of calcite (CaCO_3_). Due to the low combustion temperature in a fluidized bed and the short time of fuel and sorbent exposure to high temperature, the un-combusted calcium oxide is highly reactive [24]. Brown coal fly ash (“Turow” Power Plant) also contains metakaolinite [25]. Owing to the content of amorphous phases and grains with developed specific surface area, the hydration of fluidized-bed coal ash leads to the formation of a gel that additionally tightens the structure and improves the strength of composites with this fly ash as the curing time passes. Fluidized-bed brown coal ashes contain silicate and calcium ions that participate in the formation of phases such as ettringite and hydrated calcium silicates. The free CaO and anhydrite content activates the hydration of composites with blast furnace slag.

### 2.3. Ash from the Thermal Treatment of Municipal Sewage Sludge

Fly ash from the thermal treatment of municipal sewage sludge (TTMSS) is produced as a result of municipal sewage sludge combustion, usually in fluidized-bed boilers, at a temperature of 600–900 °C. It exhibits a relatively high content of phosphorus [26,27] and certain heavy metals [28], as well as properties that hinder their application within concrete technology, i.e., high water demand and fineness, as well as low hydraulic and pozzolanic activity [29]. This waste product is characterized by low amorphism. Quartz is the dominant phase in the ash, which also contains a group of calcium (including anhydrite) and phosphate minerals.

There have been known attempts to use TTMSS ash for the production of phosphorous fertilizers [30], for wastewater treatment (after pre-processing) [31], as a component in mineral binders [32], in concrete [33], in ceramic materials [34], and as a lightweight aggregate [35]. Some of these applications require changing the properties or pre-treatment of the ash.

One possible use of TTMSS fly ash in its raw form that does not increase its economic value, is in hardening slurries. These, owing to their properties (particularly high water content in the composition and minor requirements in terms of strength), should be resistant to the effects of this waste that are negative from the concrete technology perspective [15]. The TTMSS fly ash used in the experiment originated from the municipal sewage treatment plant in Warsaw.

### 2.4. Blast Furnace Slag

Ground granulated blast furnace slag is a hydraulic additive that partially replaces the Portland clinker used in the production of common cement. These actions result from the necessity to limit the manufacturing costs of Portland cement, as well as to reduce CO_2_ emissions to the atmosphere, and the need to maintain the requirements of the circular economy and zero-waste economy. Blast furnace cement, which contains from 35% to 95% of blast furnace slag (CEM III/A, B and C), is used whenever it is required to limit binder hydration heat and maintain the design concrete strength class. This type of cement is characterized by large strength increments over long hardening periods, and its composites exhibit a lower number of capillary pores, as well as higher structural tightness, hence, limited permeability and penetration of aggressive substances.

Blast furnace slag is a material with latent hydraulic properties that does not bind with water or binds very slowly. The slag hydration process depends on the glassy phase, an appropriate degree of grain grinding and their activation. Slag is activated by adding basic or weak acidic compounds, or by applying elevated pressure and temperature. The necessity to apply blast furnace slag binding reaction activators results from the fact that a poorly permeable layer of aluminum silicates is formed on the surfaces of grains when in contact with water. This limits water penetration inside the grains, hence stopping the hydration process. The reaction of slag binding in slurries (D, E, F, and G) was activated by adding only fluidized-bed brown coal combustion fly ash—a material with a high alkaline reaction (the pH of an aqueous solution with the ash is 12.7), characterized by appropriate hydraulic and pozzolanic properties.

### 2.5. Research Scope and Methodology

All of the hardening slurry recipes were tested for technological (in liquid state) and performance (after hardening) properties. The following were determined in the liquid state:bulk density using a Baroid scale, a simple measurement involving filling a container of known volume with the slurry and balancing the scale [36];relative viscosity using a flow viscometer, measured as the outflow time for a specified volume of slurry from a standard vessel (Marsh funnel) [36]—the longer the time, the higher the viscosity;daily water bleed, the relative volume of water released from the slurry spontaneously after being left to rest for 24 h and observed in a transparent graduated cylinder [37];structural (gel) strength, measured using a shearometer and based on the sinking depth of a hollow tin cylinder under its own weight in the slurry, after letting the slurry rest for 10 min [36].

The following was tested after the hardening slurry was cured in tap water for a specified period (depending on the additive):compressive strength via an unconfined compression test in a hydraulic press, as per [38], conducted on cylinder specimens (d = 8 cm, h = 8 cm);hydraulic conductivity (filtration coefficient k_10_, i.e., at a temperature of +10 °C), with a variable hydraulic gradient [10]. The method consists of determining, in established times, the values of water pressure in a supply tube of a certain cross-section area during the liquid’s flow through the specimen of a certain height;corrosion resistance (only slurries A and B) using the so-called adequacy test, which means a long-term hydraulic permeability test of the slurry under conditions of filtration flow of eluates from a specific landfill (Table 3) that is to be surrounded by a cut-off wall. The test, involving eluates with a complex chemical composition, does not only verify the sealing properties of a cut-off wall material—rather, it is a prognosis of its durability under conditions of complex corrosive aggressiveness. The italics in Column 5 of Table 3 mark values that exceed the permissible—in this case, those determined in the perspective of domestic regulations on protecting waters against pollutants. Among these highlighted values, a bold font additionally indicates the content of ammonia nitrogen—since the XA3 exposure class limit value according to [39], which is a requirement specified from the perspective of ammonia aggressiveness of cement-based materials—has been exceeded almost sevenfold;heavy metal leachability (only slurry C) for this purpose, hardening slurry specimens after 28 days of curing in tap water were filtered with distilled water (simulation of slurry operation in the ground). The medium flow was laminar and a quasi-constant hydraulic gradient was applied. The eluate (specimen filtrate) was collected in 7 fractions with increasing volume (associated with specimen dry mass), until the obtained L/S (liquid to solid) ratio = 10 dm^3^/kg d.m. The reaction and specific conductivity were determined in eluates obtained this way, and after they were preserved with nitric acid (V), the heavy metal content was then determined utilizing the flame atomic absorption spectroscopy (FAAS) method. The release intensity for the elements in question was calculated similarly to eluate sampling within the percolation method [40];porosity (only slurries A and B) with the mercury porosimetry method;phase composition with methods such as X-ray diffraction analysis (XRD, Bruker, Karlsruhe, Germany)—employing a Bruker D8 Advance device equipped with a LYNXEYE position-sensitive detector that operates within the Bragg-Brentano geometry using CuKα (λ = 0.15418 nm) radiation with a nickel filter—and scanning electron microscopy (SEM, Oxford Instruments Ltd., Abingdon, UK) with a ZEISS LEO 1430 scanning electron microscope equipped with an Oxford ISIS 300 energy-dispersive detector (EDS, Oxford Instruments, Abingdon, UK) at the Institute of High Pressure of the Polish Academy of Sciences (PAN) in Warsaw.

## 3. Results and Analysis

The basic properties of all slurries (A to G), in liquid state and after hardening, are shown in Table 4.

### 3.1. Slurry A and B

Figure 2 illustrates the results of long-term hydraulic conductivity tests of slurries A and B after 28 days of curing in water. The results of reference specimens subjected to tap water filtration are also shown for comparison purposes. Eluate and tap water filtration lasted continuously for 210 days. Figure 2 and the *k_10_* trend values it shows indicate the process of structure sealing (reduced hydraulic conductivity) in the material of slurry with added fluidized-bed brown coal ash B, when subjected to filtration of landfill eluates. The values in the hydraulic conductivity of this slurry were recorded both at the beginning and at the end of the test period, relative to slurry B, when penetrated by tap water.

An almost constant level of hydraulic conductivity (with a minor initial decline) throughout the entire test period can be observed in the case of a slurry with added fluidized-bed hard coal fly ash A, on being subjected to eluate filtration. The obtained results are consistent with the *k_10_* values and the variation trend for the same slurry, but with tap water filtration. Additional sealing of slurry B (during the entire measurement period) can be related with brown coal ash reactivity. This includes being in contact with substances contained in eluates. In order to identify this process, the slurries were subjected to special testing for porosity (Table 5) and phase composition (Figure 3 and Figure 4).

The parameters characterizing the microstructure of slurries subjected to tap water filtration are higher than in the case of eluate filtration, leading to reduced total porosity and maximum pore diameter. Slurry A exhibits the relatively highest reduction in porosity in terms of micropores (*v_p_* < 0.2 μm)—by 29.4%. In terms of mesopores (*v_p_* > 0.2 μm), a higher decline was recorded for slurry B—by 17.2%. The decline in the maximum pore diameter (by 42.9%) has to be recognized as the factor determining the reduced permeability of slurry B with eluate filtration, since it is the pores that have the greatest ability to generate passage discontinuities and the highest transport capacity.

Figure 3 shows the XRD diffraction patterns that indicate characteristic phases in slurries subjected to eluate and tap water filtration. Higher CaCO_3_ contents were recorded in both slurry types (A and B) after exposure to landfill eluates. This indicates a faster formation of calcium carbonate in the event of an action by a chemically complex environment, probably also due to higher supply of reactants. The greater amount of calcite (more intensive diffraction pattern peaks), especially in slurry B after eluate filtration, can explain the tighter structure of this slurry, relative to specimens filtered by tap water.

Figure 4 shows selected microstructural images of the tested hardening slurries. Their analysis indicates the higher structural tightness of slurries after eluate filtration and confirms the conclusions from previous research.

### 3.2. Slurry C

The composition of the bentonite-cement-ash slurry C contained fly ash from the thermal treatment of sewage sludge (TTMSS). This is characterized, among others, by a higher heavy metal content (Table 6). Because this leads to difficulties in the safe utilization of this ash, the research involving slurry C focused on its immobilization capacities towards selected heavy metals introduced to the slurry by TTSMM ash and (in significantly smaller amounts) by cement binder. The studies addressed the leachability of zinc (Zn), copper (Cu), lead (Pb), cadmium (Cd) and chromium (Cr) from the hardened slurry as a result of water filtration.

Nine specimens of the hardening slurry after 28 days of curing in tap water were used to study heavy metal leachability. A brief description of the applied proprietary leachability test method that takes into account the operating conditions of the slurry in a cut-off wall is presented in Table 6, and is expanded on in detail in [42].

Table 7 shows the heavy metal concentrations in individual fractions of eluates (filtrates) specimens in the course of water filtration through three exemplary specimens of slurry C. Figure 5 and Figure 6 list the changes of pH and specific conductivity of eluates as a function of cumulative L/S ratio. Table 8 presents the heavy metal immobilization level in the hardening slurry, understood as the residual amount of an element in a specimen after its leaching process that is relative to the original content in the specimen, and displayed as a mass ratio.

The high pH of eluates (Figure 5) results from the presence of cement hydration products in the hardening slurry and is similar to the pH of a porous liquid in cement matrices. The test results indicate a statistically significant relationship between the pH and the cumulative L/S ratio (ρc = −0.706). A significant partial correlation between specific conductivity and cumulative L/S ratio (significant relationship ρc = −0.723) was also observed (Figure 5). The gradual stabilization of eluate parameters indicates the possibility of the studied material reaching a state of equilibrium when in a real work environment.

Zinc concentration in eluates indicated a significant partial correlation with pH (ρc = 0.592) and cumulative L/S ratio (ρc = 0.444). Its high immobilization level (Table 8) could have resulted from a significant share of the residual fraction [43] in the total metal content in the slurry [44].

In the case of all specimen eluates, copper was leached below the determination limit (0.02 mg/dm^3^). Ash-originating copper compounds are probably in a form characterized by low mobility (element immobilization of >99.90%—Table 8), as confirmed by the results of other researchers [45,46,47], which is why it should not pose a threat to the environment, despite the high in ash concentration.

No significant partial correlation exists between lead concentration and eluate pH (probably due to the pH range of specimen eluates being too narrow) and cumulative L/S ratio. The high level of metal immobilization, similar to zinc, can evidence a high content of the residual fraction in the total content of the element in ash [44].

Cadmium concentration in most eluate fractions was below the determination limit (0.01 mg/dm^3^), which points to high immobilization of this element in the slurry. Chromium concentration, similar to zinc, indicated a statistically significant dependence on eluate pH (ρc = 0.314) and cumulative L/S ratio (ρc = −0.294). Element immobilization level in the slurry should be considered as high.

### 3.3. Slurries D–G

Bentonite-slag-ash-aqueous hardening slurries D, E, F, and G, with compositions listed in Table 1, were subjected to liquid state tests of such properties as density, viscosity, daily water bleed, and post-hardening tests after four curing periods of compressive strength and the filtration coefficient. The results are given in Table 4. Slurry microstructure was tested after 28 and 90 days of curing using scanning electron microscopy (SEM). Specimen microstructure tests revealed the presence of phases characteristic for cement composites, namely, the C-S-H phase in the form of crumpled foils, as well as ettringite in the form of hexagonal pistils (Figure 6).

Figure 7 shows changes in the compressive strength of slurries D–G over time. It can be seen that the slag dosing has the greatest impact on slurry strength. This confirms generally known trends [48]. Given the most common expectations in terms of compressive strength ranging from 0.5–1.0 MPa, the slurry following recipe E, with a compressive strength after 28 days of curing amounting to 0.52 MPa, was the most promising.

Figure 8 displays changes in the hydraulic permeability of slurries D–G over time. The analysis of slurry hydraulic conductivity (Table 4 and Figure 8) as a function of their composition (Table 1) enables the conclusion that the tightness of a hardening slurry structure increases with increasing slag dosage. This is accompanied by reduced hydraulic conductivity (filtration coefficient). The filtration coefficient in all hardening slurry recipes declines with elapsing specimen hardening time.

Considering the permeability test results in the context of requirements most common for structures intended for cut-off walls (*k_10_* ≤ 10^−9^ m/s), it can be noticed that these expectations are met by slurries E, F, and G after 28 days of curing. Combined with the strength-related requirements (multi-criteria material selection), all expectations are satisfied by slurry E.

## 4. Potential Significance of Hardening Slurries in Cut-Off Walls as a Deposit for Combustion By-Products in Terms of Circular Economy

Regardless of the detailed conclusions drawn based on individual slurries, it can be concluded that each of these is able to accumulate up to several hundred kilograms of CBPs and/or blast furnace slag in one cubic meter, without losing their properties that enabled incorporation into a single-phase cut-off wall (appropriate viscosity), while maintaining the expected strength parameters, tightness, and durability under conditions of complex environmental actions.

If we assume that cut-off wall execution can follow the two-phase method (a replacement or supplementation of the working slurry in the excavation), the doses of components of a by-product nature (power or thermal waste treatment ashes, slag) could be even greater. This would substantially increase the possibilities to deposit these materials in useful cut-off walls. The specimen estimate of these capabilities in Polish conditions is as follows.

There are approximately 8600 km of levees in Poland. According to the services responsible for the technical condition of these structures, approximately half of them pose a potential threat (mainly due to advanced age), and approximately 11% require immediate renovation. Assuming the average sizes of cut-off walls (8 m high, 0.5 m thick) and the fact that they would be built along 11% to 50% of the levee length, the volume of required hardening slurries can be estimated at 3.4 to 17.2 MM m^3^.

If only ashes of various origin were used to produce this volume of slurries, and they would be dosed at an average rate of 300 kg/m^3^ of slurry (this value, in the light of presented studies and possible cut-off wall execution technologies shall be treated as the lower limit), then the cut-off walls in levees could incorporate at least 1 to 5.2 MM tonnes of CBPs, including the ones most difficult to dispose of, such as fluidized-bed CBPs.

It is worthwhile to compare the estimated possibilities of incorporating CBPs into cut-off walls in levees with the amount of CBPs generated annually within the Polish power system. Approximately 15.6 MM tons of coal combustion slags and ashes, including approximately 2.6 MM through processes associated with calcium-based desulphurization methods (including fluidized-bed combustion), were generated in Poland in 2018 [13].

The presented test results for a specific construction material, namely hardening slurries, clearly indicate their ability to bind combustion by-products, including coal, sewage sludge, and metallurgical waste. This is possible without deterioration in their technological and performance properties after hardening. Due to the composition of certain CBPs (e.g., ones with heavy metals) potentially harmful to the environment, slurries can be a safe method for their deposition since they exhibit significant immobilization capacities. Geoengineering is the obvious field of application for hardening slurries. This includes the various execution technologies for cut-off walls, such as the ones tolerating an increased share of solids in the slurry. This amplifies the possibility to deposit even greater amounts of CBPs in the cut-off walls, as well as other components otherwise difficult to dispose of.

Hardening slurries and their wide application possibilities can, therefore, be an effective and attractive form of depositing significant CBP amounts, including fluidized-bed CBPs that are difficult to manage otherwise. Hence, the described direction for the utilization of these materials can successfully contribute to the implementation of the circular economy concept.

## 5. Conclusions

Cement-bentonite-ash-aqueous slurries subjected to landfill eluate filtration did not undergo corrosive destruction. Instead, they exhibited an increasing tightness (reduced hydraulic conductivity coefficient) compared to the permeability tests utilizing tap water. The test results confirm the high resistance of slurries with fluidized-bed fly ash additives (both hard and brown coal) to an aggressive environment of municipal landfill eluates.It is possible to produce a hardening slurry with desired technological and performance properties using fly ash from Thermal Treatment of Municipal Sewage Sludge (TTMSS).The hardening slurry with TTMSS fly ash was characterized by high immobilization in respect of tested heavy metals.The most leachable element was lead. The concentrations of other studied elements (zinc, copper, cadmium, and mercury) in eluates were low (below the determination limit).It is possible to produce cement-free bentonite-aqueous hardening slurries with blast furnace slag activated by the addition of specific fluidized-bed brown coal combustion fly ash as the binder. The progress of binder hydration over time is evidenced by the changes ongoing in the course of slurry curing. They include increasing compressive strength and declining hydraulic conductivity, and result in a tightening of the slurry structure.

## Figures and Tables

**Figure 1 materials-14-02104-f001:**
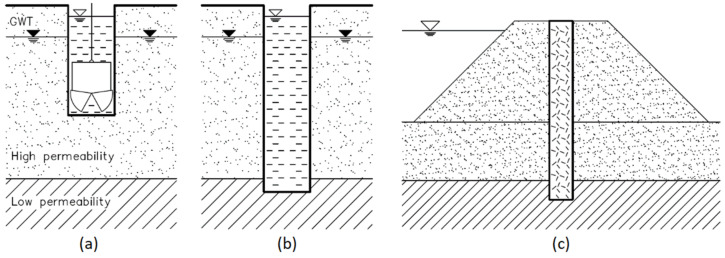
Single-phase execution schematics of a cut-off wall utilizing a hardening slurry: (**a**) deepening of a narrow linear excavation with walls supported by the slurry (added as work progresses); (**b**) excavation at a desired depth, awaiting the bonding of slurry binder; (**c**) sample operating conditions of a hardened slurry in a cut-off wall of an embankment. GWT—groundwater table.

**Figure 2 materials-14-02104-f002:**
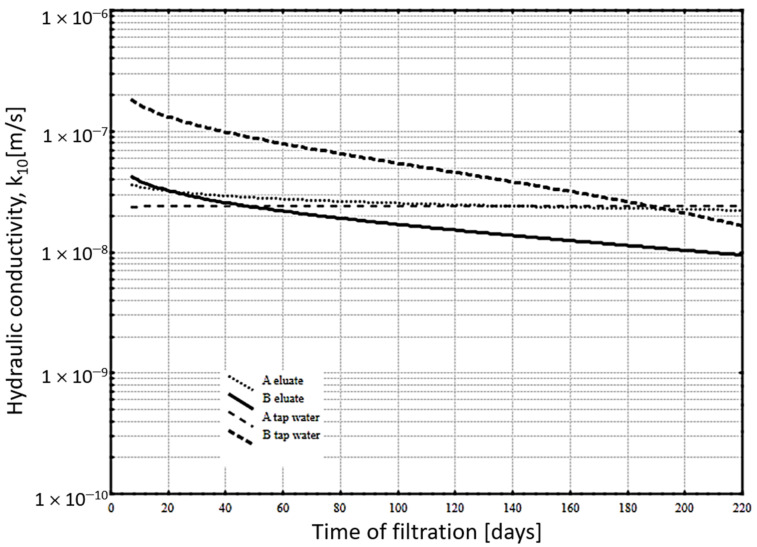
Hydraulic conductivity of hardening slurries with addition of fluidized-bed hard coal A and brown coal B fly ash-as a function of time (trend lines).

**Figure 3 materials-14-02104-f003:**
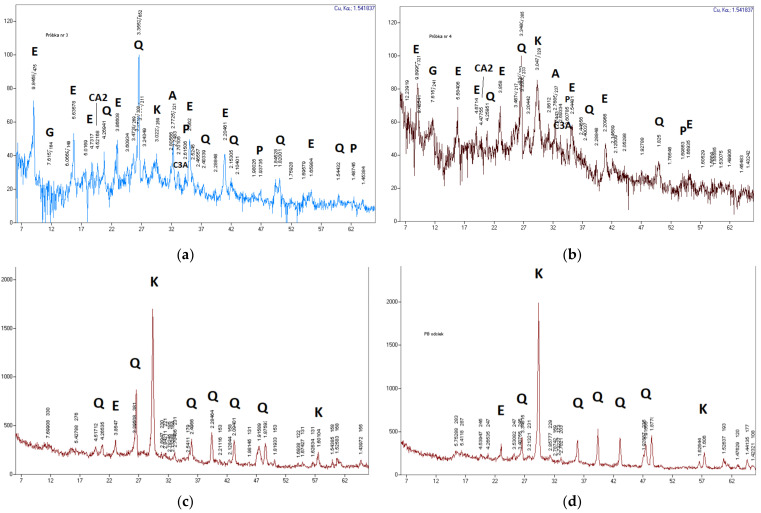
Diffraction patterns (XRD) of hardening slurry specimens A and B: (**a**) slurry A after tap water filtration; (**b**) slurry B after tap water filtration; (**c**) slurry A after landfill eluate filtration; (**d**) slurry B after landfill eluate filtration. Designations: A—alite, C3A—tricalcium aluminate, CA2—grossite, E—ettringite, K—calcite, Q—quartz, P—portlandite, G—gypsum.

**Figure 4 materials-14-02104-f004:**
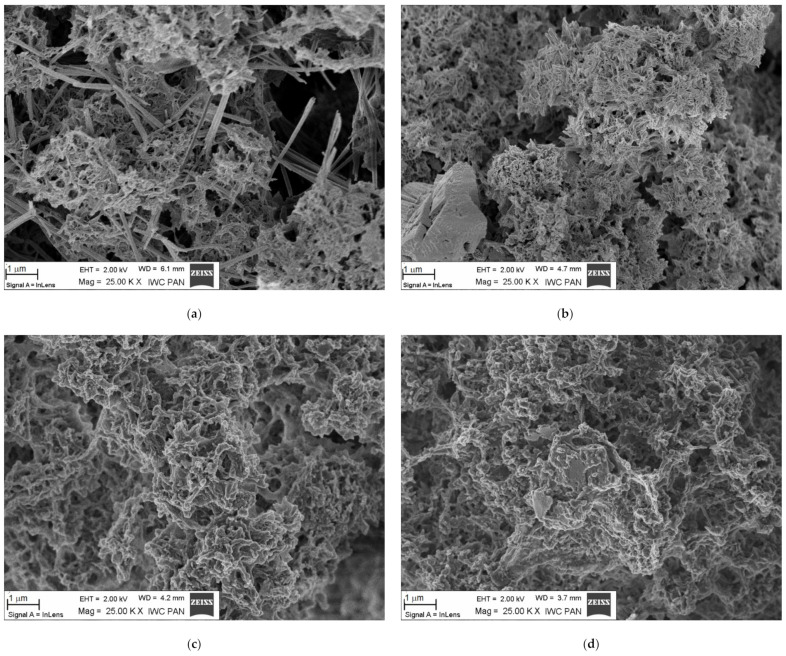
The microstructure of the hardening slurry specimens (SEM). (**a**) Slurry A after tap water filtration: C-S-H, ettringite cluster; (**b**) slurry B after tap water filtration: C-S-H and clinker relicts, pores; (**c**) slurry A after landfill eluate filtration: C-S-H and pores; (**d**) slurry B after landfill eluate filtration: C-S-H, clinker relict, pores.

**Figure 5 materials-14-02104-f005:**
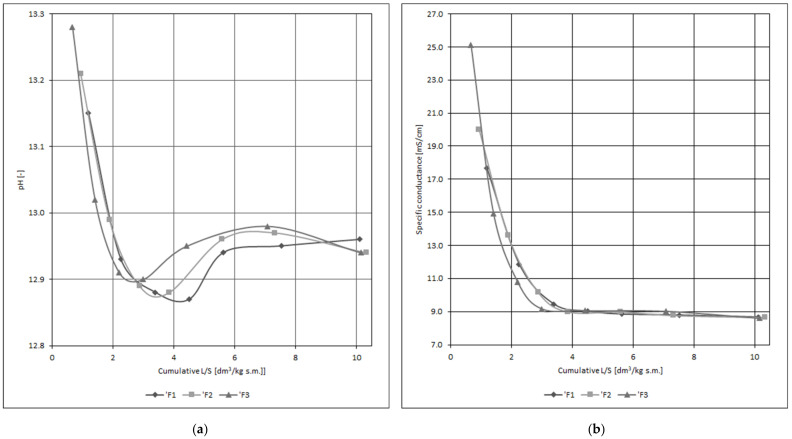
Change of eluate properties depending on cumulative L/S ratio, hardening slurry: (**a**) change of pH; (**b**) change of specific conductance.

**Figure 6 materials-14-02104-f006:**
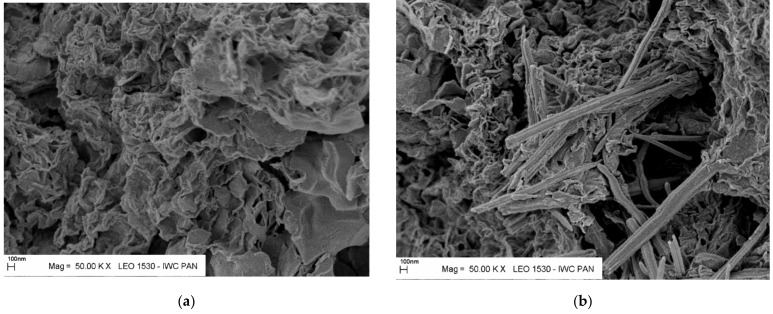
The microstructure of slurry E specimens after 28 (**a**) and 90 (**b**) days of curing; visible C-S-H phase and ettringite.

**Figure 7 materials-14-02104-f007:**
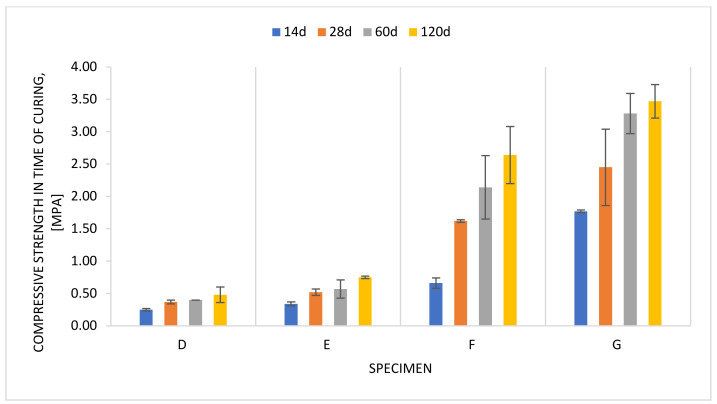
Compressive strength of slurries D–G in the course of curing.

**Figure 8 materials-14-02104-f008:**
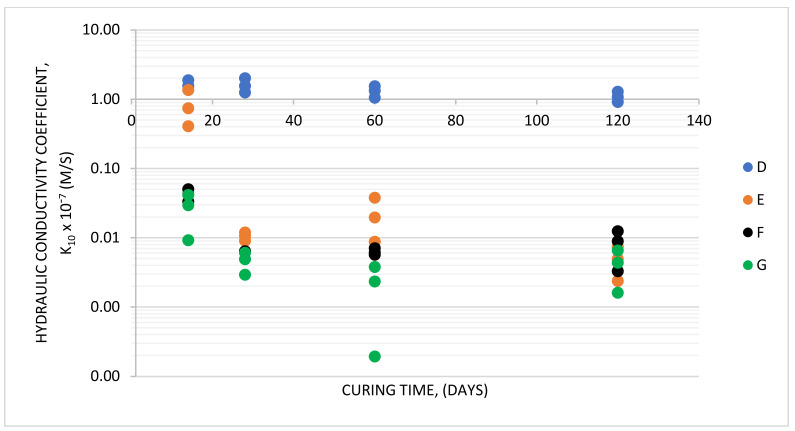
Hydraulic conductivity of slurries D–G in the course of curing.

**Table 1 materials-14-02104-t001:** Compositions of tested hardening slurries.

Component	Content of Component [kg/m^3^] in the Slurry:						
	A	B	C	D	E	F	G
Tap water	841	842	841	879	929	869	807
Sodium bentonite	34	25	21	35	56	35	48
CEM I 32.5 R cement	137	143	380	-	-	-	-
Ground blast furnace slag	-	-	-	171	181	413	383
Hard coal fly ash-fluidized bed	272	-	-	-	-	-	-
Brown coal fly ash-fluidized bed	-	275	-	154	14	13	141
TTMSS fly ash ^1^	-	-	84	-	-	-	-

^1^ Thermal Treatment of Municipal Sewage Sludge (TTMSS).

**Table 2 materials-14-02104-t002:** Oxide composition in CBPs and blast furnace slag used in hardening slurry tests.

No.	Chemical Composition/Physical Properties	Ingredient Content [mass %] in:				
		Fluidized-Bed Hard Coal Fly Ash	Fluidized-Bed Brown Coal Fly Ash ^1^		TTMSS Fly Ash	Ground Granular Blast Furnace Slag
		in Slurry A	in Slurry B	in Slurries D–G	in Slurry C	Slurries D–G
1	Cl	0.070	0.015	0.040	0.038	0.110
2	SO_3_	6.19	3.33	5.66	2.78	0.67
3	CaO	12.52	17.38	21.10	13.2	45.4
4	CaO_free_	3.89	6.27	7.00	0.12	0.80
5	SiO_2_	35.73	36.51	34.87	36.4	36.5
6	Al_2_O_3_	19.89	27.10	23.09	18.1	9.20
7	Fe_2_O_3_	6.30	5.91	4.82	5.7	1.56
8	Na_2_O	0.82	1.58	0.02	2.26	0.01
9	K_2_O	1.88	0.74	0.01	2.95	0.05
10	MgO	2.20	2.31	1.69	4.15	5.89
11	Loss on ignition	13.65	3.35	3.15	2.09	0.01

^1^ fly ash originated from two different deliveries.

**Table 3 materials-14-02104-t003:** Physical and chemical parameters of municipal landfill eluates and values permitted by legal regulations [41].

No.	Index	Unit	Permissible Index Value	Index Value in Eluate
1	pH	-	6.5–9.0	7.71
2	BOD_5_	mg O_2_/dm^3^	25–50	4850
3	COD	mg O_2_/dm^3^	125–250	13,136
4	Total nitrogen (*N*)	mg *N*/dm^3^	30	884
5	Ammonium nitrogen (NH_4_)	mg NH_4_/dm^3^	10–30	763
6	Total phosphorus (*p*)	mg *p*/dm^3^	2–10	78.4
7	Chlorides (Cl)	mg/dm^3^	1000	1597
8	Sulphates (SO_4_)	mg/dm^3^	500	37.5
9	∑ Cl, SO_4_	mg/dm^3^	1500	1634.5
10	Lead (Pb)	mg Pb/dm^3^	0.1–0.5	4.25
11	Copper (Cu)	mg Cu/dm^3^	0.1–0.5	0.336
12	Zinc (Zn)	mg Zn/dm^3^	2.0	0.808
13	Nickel (Ni)	mg Ni/dm^3^	0.1–0.5	0.394
14	Total suspended solids	mg/dm^3^	35–70	6774

**Table 4 materials-14-02104-t004:** Basic properties of tested hardening slurries in liquid and solid state.

Property or Parameter	Slurry Designation:						
	A	B	C	D	E	F	G
w/c ^1^	6.13	5.88	2.21	-	-	-	-
w/d ^2^	1.90	1.90	1.73	14.6	26.1	4.4	3.9
Density [kg/m^3^]	1.29	1.30	1.33	1.24	1.18	1.33	1.38
Relative viscosity [s]	45	39	50	38	40	37	46
Structural strength of gel after 10 min of standing time [Pa]	-	-	5.2	3.0	8.0	3.8	40.0
Daily water bleed [%]	3.0	5.0	3.9	8.5	1.5	4.0	2.0
Compressive strength [MPa] after: –14 days	-	-	-	0.25	0.34	0.66	1.77
–28 days	1.21	1.38	1.80	0.37	0.52	1.62	2.45
–60 days	-	-	-	0.40	0.57	2.14	3.28
–120 days	-	-	-	0.48	0.75	2.64	3.47
Hydraulic conductivity k_10_ [m/s] after: –14 days	-	-	-	1.56 × 10^−7^	7.48 × 10^−8^	4.23 × 10^−9^	2.98 × 10^−9^
–28 days	3.5 × 10^−8^	2.5 × 10^−8^	9.6 × 10^−9^	1.56 × 10^−7^	1.07 × 10^−9^	6.35 × 10^−10^	4.95 × 10^−10^
–60 days	-	-	-	1.34 × 10^−7^	1.98 × 10^−9^	6.19 × 10^−10^	2.35 × 10^−10^
–120 days	-	-	-	1.09 × 10^−7^	4.99 × 10^−10^	9.00 × 10^−10^	4.40 × 10^−10^

^1^ mixing water to cement mass ratio; ^2^ mixing water to dry ingredient mass ratio.

**Table 5 materials-14-02104-t005:** List of parameters characterizing the microstructure of tested hardening slurries.

No.	Type of Solution	A	B	A	B	A	B	A	B	A	B
		A_p_ [m^2^/g]		v_p_ > 0.2 μm		v_p_ < 0.2 μm		P_c_ [%]		d_max_ [μm]	
1	Tap water	99.54	100.91	0.89	0.99	0.46	0.44	76.22	75.90	4.0	7.0
2	Eluate	94.45	85.76	0.79	0.82	0.32	0.39	73.88	72.13	3.0	4.0
3	2:1 [%]	94.9	85.0	88.8	82.8	69.6	88.6	96.9	95.0	75.0	57.1

Where: A_p_—total porous area, [m^2^/g]; v_p_ < 0.2—volume of pores with diameters below 0.2, in mL/g; v_p_ > 0.2—volume of pores with diameters above 0.2 μm, in mL/g; P_c_—total specimen porosity, [%]; d_max_—maximum pore diameter, [μm].

**Table 6 materials-14-02104-t006:** Content of selected heavy metals in a hardening slurry and its components.

No.	Metal	TTMSS Fly Ash	CEM I 32.5 R	Hardening Slurry
		Value [mg/kg d.m.]		
1	Zinc (Zn)	3290 ± 83	804 ± 24	1057 ± 66
2	Copper (Cu)	808 ± 24	120 ± 4	206 ± 14
3	Lead (Pb)	83.0 ± 6.5	104 ± 7	84.5 ± 8.8
4	Cadmium (Cd)	14.0 ± 0.7	10.8 ± 0.6	9.6 ± 0.9
5	Chromium (Cr)	179 ± 9	64.4 ± 4.2	71.8 ± 6.8

**Table 7 materials-14-02104-t007:** Heavy metal concentrations in eluates.

Name	Eluate	Concentration									
		Zinc		Copper		Lead		Cadmium		Chromium	
		C_i_ [mg/dm^3^]	Δ [%]	C_i_ [mg/dm^3^]	Δ [%]	C_i_ [mg/dm^3^]	Δ [%]	C_i_ [mg/dm^3^]	Δ [%]	C_i_ [mg/dm^3^]	Δ [%]
F1	1	0.022	29	<0.02	-	0.132	24.2	<0.01	-	0.046	27
	2	<0.01	-	<0.02	-	0.121	14.5	<0.01	-	0.046	17.7
	3	0.02	50	<0.02	-	0.166	15.5	<0.01	-	0.04	2.9
	4	<0.01	-	<0.02	-	0.244	1.8	<0.01	-	0.04	9.1
	5	<0.01	-	<0.02	-	0.12	17.8	<0.01	-	0.088	14.4
	6	<0.01	-	<0.02	-	0.123	10.5	<0.01	-	0.04	25.8
	7	<0.01	-	<0.02	-	0.122	14.1	<0.01	-	0.052	6
F2	1	<0.01	-	<0.02	-	0.133	12.5	<0.01	-	0.064	5.1
	2	0.021	39	<0.02	-	0.149	8.2	<0.01	-	0.044	16.6
	3	0.038	36	<0.02	-	0.193	4.1	<0.01	-	0.045	33
	4	<0.01	-	<0.02	-	0.103	13.3	<0.01	-	0.052	19.4
	5	<0.01	-	<0.02	-	0.129	6.6	0.014	29.1	0.068	13.9
	6	<0.01	-	<0.02	-	0.129	14.5	<0.01	-	<0.03	-
	7	<0.01	-	<0.02	-	0.119	1.9	<0.01	-	0.056	12.7
F3	1	<0.01	-	<0.02	-	0.143	10.6	<0.01	-	0.085	3.5
	2	0.016	119	<0.02	-	0.099	11.3	<0.01	-	0.052	9.7
	3	0.014	51	<0.02	-	0.196	15.3	<0.01	-	0.042	19.2
	4	<0.01	-	<0.02	-	0.104	7.3	<0.01	-	0.031	25.6
	5	<0.01	-	<0.02	-	0.122	7.7	0.011	37.2	<0.03	-
	6	<0.01	-	<0.02	-	0.118	20.5	<0.01	-	<0.03	-
	7	<0.01	-	<0.02	-	0.1	19.1	<0.01	-	0.035	11

**Table 8 materials-14-02104-t008:** Heavy metal immobilization.

Sample Name	Zinc [%]	Copper [%]	Lead [%]	Cadmium [%]	Chromium [%]
F1	>99.99	>99.90	98.31	>98.95	99.30
F2	>99.99	>99.90	98.38	>98.85	>99.25
F3	>99.99	>99.90	98.57	>98.93	>99.47

## Data Availability

The data presented in this study are available on request from the corresponding author.

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
