# Peer review of "Hardening Slurries with Fluidized-Bed Combustion By-Products and Their Potential Significance in Terms of Circular Economy"

_materials, 2021, doi:10.3390/ma14092104_

Round 1
Reviewer 1 Report
The manuscript entitled "Hardening slurries with fluidized-bed combustion by-products and their significance in terms of circular economy” investigated the effect of the addition of by-products of fluidized-bed combustion of hard, brown coal and municipal sewage sludge, as well as ground granulated blast furnace slag (a by-product of steel industry) on the properties of slurries, in the liquid state and after hardening.
The manuscript lacks clarity and should not be accepted in this current condition. This reviewer recommends major editing.
Comments:
- The title of the manuscript includes "their significance in terms of circular economy". However, this reviewer did not find any discussion related to this significant effect except in lines 232 and 233 which is not enough.
- The manuscript could benefit greatly from professional editing to improve technical writing and English.
- Why the authors wrote sentences in very short paragraphs throughout the manuscript?
- The methodology used in this investigation should be highlighted in the abstract.
- This reviewer has a big question mark about section 1.1. The introduction section should include the literature review to highlight all related previous work as well as the significance of the current work. I am not sure what is the main purpose of providing that section “research subject”?
- Table 1: I think the letters A, B, C, D, E, F, and G represents different recipe. Is that correct? If so, the authors should address this before Table 1.
- Section 2.5: The authors did not mention what are the shapes of the tested specimens that used for the hardening properties?
- Line 301: The verb "lists" is more suitable for Tables. However, figures should use "illustrates, shows, or presents".
- Line 304: Figure 2 shows results for 220 days. Why?
- Line 305: This sentence should be corrected.
- Table 4: Are there reasons that the compressive strength was measured at an age of 60-day and the hydraulic conductivity at 90-day age?
- Line 372: I think it should be Figure 5 only which is Fig. 5(a) and Fig. 5(b).
- Figure 6: The caption of this Figure should be corrected.
- I think the fifth point, in conclusion, should be included in the conclusion points or the discussion section. No need to add a summary section after the authors presented the conclusion points.
Author Response
Please see the attachement.

Reviewer 2 Report
The paper concerns the study of hardening slurries composed of fly ash (from combustion and thermal treatment of municipal sewage sludge) and blast furnace slag (from steel industry), referred to in the paper as combustion by-products, to be used as cut-off walls. The paper is based on laboratory research on the hardening slurries' main properties.
The paper review results in the following main comments suggested to the authors to improve and clarify the text manuscript:
- The paper is not original, but some innovation is found regarding the type of by-products used in the hardening slurry composition. The authors are invited to point out the paper's innovation by complementing the introduction section with the literature review on the state-of-the-art of paper subject.
- In section 2, I recommend to the authors presenting the standard tests in the manuscript text. In the case of European’s standards, the references should give the titles in English.
- Table 7 is presented before its reference in the manuscript test. Please, confirm that it is correct.
- Horizontal axles of Figures 7 and 8 represent the curing time in days. However, the scale is not in correspondence to this unit (days). So in the particular case of Figure 8, the authors should explain the objective of the trend lines in more detail. Is the representation of these lines correct, taking into consideration the horizontal scale? Besides, the correlation coefficient is not presented for these power lines of the hydraulic conductivity.
- Finally, the paper's title pretends to establish the significance of the subject – hardening slurries with fluidized combustion by-products – to the circular economy. However, the paper's purpose was to demonstrate the hardening slurries' feasibility with by-products in terms of some properties. It is obvious that the consumption of these by-products with this finality has advantages and contributes to the circular economy. But how is this objective developed in the paper? The present manuscript is not in correspondence to this title, and the authors should reformulate the objective of the paper.
Round 2
Reviewer 1 Report
The authors addressed all the reviewer's comments and the manuscript can be accepted for publication.
Reviewer 2 Report
The authors addressed the main suggestions of the first round of review well.